# On GANs and GMMs

**Eitan Richardson**
School of Computer Science and Engineering
The Hebrew University of Jerusalem
Jerusalem, Israel
eitanrich@cs.huji.ac.il

**Yair Weiss**
School of Computer Science and Engineering
The Hebrew University of Jerusalem
Jerusalem, Israel
yweiss@cs.huji.ac.il

## Abstract

A longstanding problem in machine learning is to find unsupervised methods that can learn the statistical structure of high dimensional signals. In recent years, GANs have gained much attention as a possible solution to the problem, and in particular have shown the ability to generate remarkably realistic high resolution sampled images. At the same time, many authors have pointed out that GANs may fail to model the full distribution ("mode collapse") and that using the learned models for anything other than generating samples may be very difficult.

In this paper, we examine the utility of GANs in learning statistical models of images by comparing them to perhaps the simplest statistical model, the Gaussian Mixture Model. First, we present a simple method to evaluate generative models based on relative proportions of samples that fall into predetermined bins. Unlike previous automatic methods for evaluating models, our method does not rely on an additional neural network nor does it require approximating intractable computations. Second, we compare the performance of GANs to GMMs trained on the same datasets. While GMMs have previously been shown to be successful in modeling small patches of images, we show how to train them on full sized images despite the high dimensionality. Our results show that GMMs can generate realistic samples (although less sharp than those of GANs) but also capture the full distribution, which GANs fail to do. Furthermore, GMMs allow efficient inference and explicit representation of the underlying statistical structure. Finally, we discuss how GMMs can be used to generate sharp images. [1]

## 1 Introduction

Natural images take up only a tiny fraction of the space of possible images. Finding a way to explicitly model the statistical structure of such images is a longstanding problem with applications to engineering and to computational neuroscience. Given the abundance of training data, this would also seem a natural problem for unsupervised learning methods and indeed many papers apply unsupervised learning to small patches of images [42, 4, 32]. Recent advances in deep learning, have also enabled unsupervised learning of full sized images using various models: Variational Auto Encoders [21, 17], PixelCNN [40, 39, 23, 38], Normalizing Flow [9, 8] and Flow GAN [14]. [2]

Perhaps the most dramatic success in modeling full images has been achieved by Generative Adversarial Networks (GANs) [13], which can learn to generate remarkably realistic samples at high resolution [34, 26], (Fig. 1). A recurring criticism of GANs, at the same time, is that while they are excellent at generating pretty pictures, they often fail to model the entire data distribution, a phenomenon usually referred to as *mode collapse*: "Because of the mode collapse problem, applications

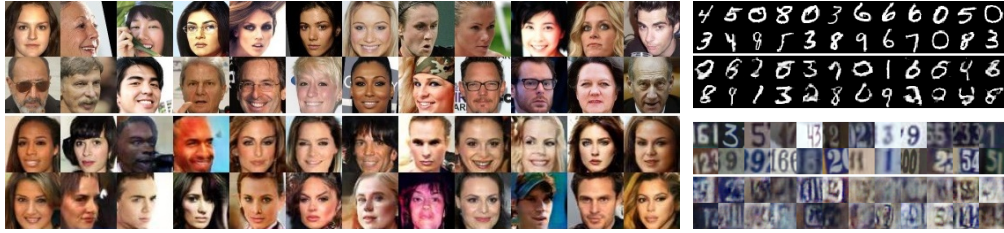

Figure 1: Samples from three datasets (first two rows) and samples generated by GANs (last two rows): CelebA - WGAN-GP, MNIST - DCGAN, SVHN - WGAN

of GANs are often limited to problems where it is acceptable for the model to produce a small number of distinct outputs" [12]. (see also [35, 29, 34, 26].) Another criticism is the lack of a robust and consistent evaluation method for GANs [18, 10, 28].

Two evaluation methods that are widely accepted [28, 1] are *Inception Score* (IS) [34] and *Fréchet Inception Distance* (FID) [16]. Both methods rely on a deep network, pre-trained for classification, to provide a low-dimensional representation of the original and generated samples that can be compared statistically. There are two significant drawbacks to this approach: the deep representation is insensitive to image properties and artifacts that the underlying classification network is trained to be invariant to [28, 18] and when the evaluated domain (e.g. faces, digits) is very different from the dataset used to train the deep representation (e.g. ImageNet) the validity of the test is questionable [10, 28].

Another family of methods are designed with the specific goal of evaluating the diversity of the generated samples, regardless of the data distribution. Two examples are applying a perceptual multi-scale similarity metric (MS-SSIM) on random patches [31] and, basing on the *Birthday Paradox* (BP), looking for the most similar pair of images in a batch [3]. While being able to detect severe cases of mode collapse, these methods do not manage (or aim) to measure how well the generator captures the true data distribution [20].

Many unsupervised learning methods are evaluated using log likelihood on held out data [42] but applying this to GANs is problematic. First, since GANs by definition only output samples on a manifold within the high dimensional space, converting them into full probability models requires an arbitrary noise model [2]. Second, calculating the log likelihood for a GAN requires integrating out the latent variable and this is intractable in high dimensions (although encouraging results have been obtained for smaller image sizes [41]). As an alternative to log likelihood, one could calculate the Wasserstein distance betweeen generated samples and the training data, but this is again intractable in high dimensions so approximations must be used [20].

Overall, the current situation is that while many authors criticize GANs for "mode collapse" and decry the lack of an objective evaluation measure, the focus of much of the current research is on improved learning procedures for GANs that will enable generating high quality images of increasing resolution, and papers often include sentences of the type "we feel the quality of the generated images is at least comparable to the best published results so far." [20].

The focus on the quality of the generated images has perhaps decreased the focus on the original question: to what extent are GANs learning useful statistical models of the data? In this paper, we try to address this question more directly by comparing GANs to perhaps the simplest statistical model, the Gaussian Mixture Model. First, we present a simple method to evaluate generative models based on relative proportions of samples that fall into predetermined bins. Unlike previous automatic methods for evaluating models, our method does not rely on an additional neural network nor does it require approximating intractable computations. Second, we compare the performance of GANs to GMMs trained on the same datasets. While GMMs have previously been shown to be successful in modeling small patches of images, we show how to train them on full sized images despite the high dimensionality. Our results show that GMMs can generate realistic samples (although less sharp than those of GANs) but also capture the full distribution which GANs fail to do. Furthermore, GMMs allow efficient inference and explicit representation of the underlying statistical structure. Finally, we discuss two methods in which sharp and realistic images can be generated with GMMs.

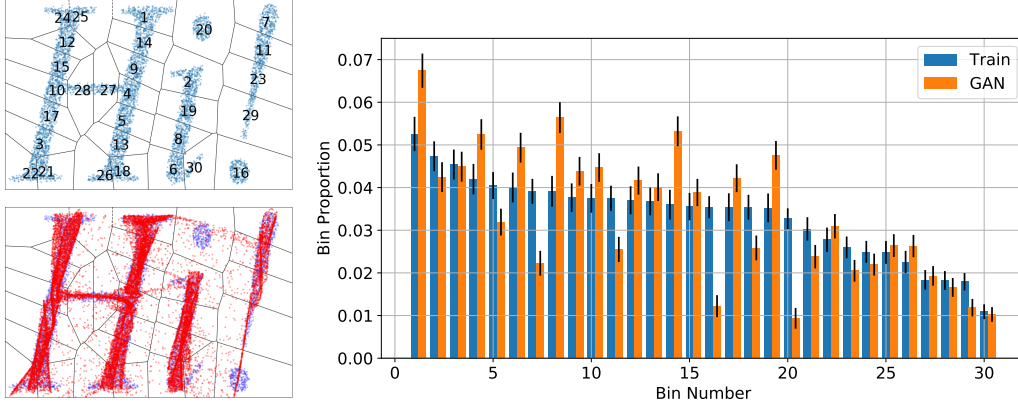

Figure 2: Our proposed evaluation method on a toy example in $\mathbb{R}^2$. Top-left: The training data (blue) and binning result - Voronoi cells (numbered by bin size). Bottom-left: Samples (red) drawn from a GAN trained on the data. Right: Comparison of bin proportions between the training data and the GAN samples. Black lines = standard error (*SE*) values.

## 2   A New Evaluation Method

Our proposed evaluation method is based on a very simple observation: If we have two sets of samples and they both represent the same distribution, then the number of samples that fall into a given bin should be the same up to sampling noise. More formally, we define $I_B(s)$ as an indicator function for bin $B$. $I_B(s) = 1$ if the sample $s$ falls into the bin $B$ and zero otherwise. Let $\{s_i^p\}$ be $N_p$ samples from distribution $p$ and $\{s_j^q\}$ be $N_q$ samples from distribution $q$, then if $p = q$, we expect $\frac{1}{N_p} \sum_i I_B(s_i^p) \approx \frac{1}{N_q} \sum_j I_B(s_j^q)$.

The decision whether the number of samples in a given bin are *statistically different* is a classic *two-sample problem* for Bernoulli variables [7]. We calculate the *pooled sample proportion $P$* (the proportion that falls into $B$ in the joined sets) and its standard error: $SE = \sqrt{P(1 - P)[1/N_p + 1/N_q]}$. The test statistic is the $z$-score: $z = \frac{P_p - P_q}{SE}$, where $P_p$ and $P_q$ are the proportions from each sample that fall into bin $B$. If the probability of the observed test statistic is smaller than a threshold (determined by the significance level) then the number is *statistically different*. There is still the question of which bin to use to compare the two distributions. In high dimensions, a randomly chosen bin in a uniform grid is almost always going to be empty. We propose to use *Voronoi cells*. This guarantees that each bin is expected to contain some samples.

Our binning-based evaluation method is demonstrated in Fig. 2, using a toy example where the data is in $\mathbb{R}^2$. We have a set of $N_p$ training samples from the reference distribution $p$ and a set of $N_q$ samples with distribution $q$, generated by the model we wish to evaluate. To define the Voronoi cells, we perform K-means clustering of the $N_p$ training samples to some arbitrary number of clusters $K$ ($K \ll N_p, N_q$). Each training sample $s_i^p$ is assigned to one of the $K$ cells (bins). We then assign each generated sample $s_j^q$ to the nearest (L2) of the $K$ centroids. We perform the two-sample test on each cell separately and report the *number of statistically-different bins* (NDB). According to the classical theory of hypothesis testing, if the two samples do come from the same distribution, then the NDB score divided by $K$ should be equal to the significance level (0.05 in our experiments).

Note that unlike the popular IS and FID, our NDB method is applied directly on the image pixels and does not rely on a representation learned for other tasks. This makes our metric domain agnostic and sensitive to different image properties the deep-representation is insensitive to. Compared to MS-SSIM and BP, our method has the advantage of providing a metric between the data and generated distributions and not just measuring the general diversity of the generated sample.

A possible concern about using Voronoi cells as bins is that this essentially treats images as vectors in pixel spaces, where L2 distance may not be meaningful. In the supplementary material we show that for the datasets we used, the bins are usually semantically meaningful. Even in cases where the bins do not correspond to semantic categories, we still expect a good generative model to preserve

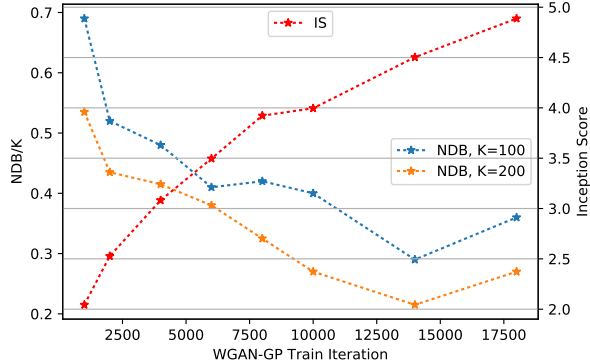

Figure 3: NDB (divided by $K$) vs Inception Score during training iterations of WGAN-GP on CIFAR-10 [24]. The two metrics correlate, except towards the end of the training, possibly indicating sensitivity to different image attributes.

the statistics of the training set. Fig. 3 demonstrates the validity of NDB to a dataset with a more complex image structure, such as CIFAR-10, by comparing it to IS.

# 3    Full Image Gaussian Mixture Model

In order to provide context on the utility of GANs in learning statistical models of images, we compare it to perhaps the simplest possible statistical model: the Gaussian Mixture Model (GMM) trained on the same datasets.

There are two possible concerns with training GMMs on full images. The first is the dimensionality. If we work with $64 \times 64$ color images then a single covariance matrix will have $7.5 \times 10^7$ free parameters and during training we will need to store and invert matrices of this size. The second concern is the complexity of the distribution. While a GMM can approximate many densities with a sufficiently large number of Gaussians, it is easy to construct densities for which the number of Gaussians required grows exponentially with the dimension.

In order to address the computational concern, we use a GMM training algorithm where the memory and complexity grow linearly with dimension (not quadratically as in the standard GMM). Specifically we use the *Mixture of Factor Analyzers* [11], as described in the next paragraph. Regarding the second concern, our experiments (section 4) show that for the tested datasets, a relatively modest number of components is sufficient to approximate the data distribution, despite the high dimensionality. Of course, this may not be necessarily true for every dataset.

Probabilistic PCA [37, 36] and Factor Analyzers [22, 11] both use a rectangular scale matrix $A_{d \times l}$ multiplying the latent vector $z$ of dimension $l \ll d$, which is sampled from a standard normal distribution. Both methods model a normal distribution on a low-dimensional subspace embedded in the full data space. For stability, isotropic (PPCA) or diagonal-covariance (Factor Analyzers) noise is added. We chose to use the more general setting of Factor Analyzers, allowing to model higher noise variance in specific pixels (for example, pixels containing mostly background).

The model for a single Factor Analyzers component is:

$$x = Az + \mu + \epsilon \, , \, z \sim \mathcal{N}(0, \, I) \, , \, \epsilon \sim \mathcal{N}(0, \, D) \, , \tag{1}$$

where $\mu$ is the mean and $\epsilon$ is the added noise with a diagonal covariance $D$. This results in the Gaussian distribution $x \sim \mathcal{N}(\mu, \, AA^T + D)$. The number of free parameters in a single Factor Analyzers component is $d(l+2)$, and $K[d(l+2)+1]$ in a Mixture of Factor Analyzers (MFA) model with $K$ components, where $d$ and $l$ are the data and latent dimensions.

## 3.1    Avoiding Inversion of Large Matrices

The log-likelihood of a set of $N$ data points in a Mixture of $K$ Factor Analyzers is:

$$\mathcal{L} = \sum_{n=1}^{N} \log \sum_{i=1}^{K} \pi_i P(x_n | \mu_i, \Sigma_i) = \sum_{n=1}^{N} \log \sum_{i=1}^{K} e^{[log(\pi_i) + \log P(x_n | \mu_i, \Sigma_i)]}, \tag{2}$$

where $\pi_i$ are the mixing coefficients. Because of the high dimensionality, we calculate the log of the normal probability and the last expression is evaluated using *log sum exp* operation over the $K$ components.

The log-probability of a data point $x$ given the component is evaluated as follows:

$$logP(x|\mu,\Sigma) = -\frac{1}{2}\big[d\log(2\pi) + \log\det(\Sigma) + (x-\mu)^T\Sigma^{-1}(x-\mu)\big] \tag{3}$$

Using the Woodbury matrix inversion lemma:

$$\Sigma^{-1} = (AA^T+D)^{-1} = D^{-1} - D^{-1}A(I+A^TD^{-1}A)^{-1}A^TD^{-1} = D^{-1} - D^{-1}AL_{l\times l}^{-1}A^TD^{-1} \tag{4}$$

To avoid storing the $d\times d$ matrix $\Sigma^{-1}$ and performing large matrix multiplications, we evaluate the Mahalanobis distance as follows (denoting $\hat{x} = (x-\mu)$):

$$\hat{x}^T\Sigma^{-1}\hat{x} = \hat{x}^T[D^{-1}-D^{-1}AL^{-1}A^TD^{-1}]\hat{x} = \hat{x}^T[D^{-1}\hat{x}-D^{-1}AL^{-1}(A^TD^{-1}\hat{x})] \tag{5}$$

The log-determinant is calculated using the matrix determinant lemma:

$$\log\det(AA^T+D) = \log\det(I+A^TD^{-1}A) + \log\det D = \log\det L_{l\times l} + \sum_{j=1}^{d}\log d_j \tag{6}$$

Using equations 4 - 6, the complexity of the log-likelihood computation is linear in the image dimension $d$, allowing to train the MFA model efficiently on full-image datasets.

Rather than using EM [22, 11] (which is problematic with large datasets) we decided to optimize the log-likelihood (equation 2) using Stochastic Gradient Descent and utilize available *differentiable programming* frameworks [1] that perform the optimization on GPU. The model is initialized by K-means clustering of the data and then Factor Analyzers parameters estimation for each component separately. The supplementary material provides additional details about the training process.

## 4   Experiments

We conduct our experiments on three popular datasets of natural images: CelebA [27] (aligned, cropped and resized to $64\times64$), SVHN [30] and MNIST [25]. On these three datasets we compare the MFA model to the following generative models: GANs (DCGAN [33], BEGAN [5] and WGAN [2]). On the more challenging CelebA dataset we also compared to WGAN-GP [15]) and Variational Auto-encoders (VAE [21], VAE-DFC [17]). We compare the GMM model to the GAN models along three attributes: (1) visual quality of samples (2) our quantitative NDB score and (3) ability to capture the statistical structure and perform efficient inference.

Random samples from our MFA models trained on the three datasets are shown in Fig. 4. Although the results are not as sharp as the GAN samples, the images look realistic and diverse. As discussed earlier, one of the concerns about GMMs is the number of components required. In the supplementary material, we show the log-likelihood of the test set and the quality of a reconstructed random test image as a function of the number of components. As can be seen, they both converge with a relatively small number of components.

We now turn to comparing the models using our proposed new evaluation metric.

We trained all models, generated 20,000 new samples and evaluated them using our evaluation method (section 2). Tables 1 - 3 present the evaluation scores for 20,000 samples from each model. We also included, for reference, the score of 20,000 samples from the training and test sets. The simple MFA model has the best (lowest) score for all values of $K$. Note that neither the bins nor the number of bins is known to any of the generative models. The evaluation result is consistent over multiple runs and is insensitive to the specific NDB clustering mechanism (e.g. replacing K-means with agglomerative clustering). In addition, initializing MFA differently (e.g. with k-subspaces or random models) makes the NDB scores slightly worse but still better than most GANs.

The results show clear evidence of mode collapse (large distortion from the train bin-proportions) in BEGAN and DCGAN and some distortion in WGAN. The improved training in WGAN-GP seems to reduce the distortion.

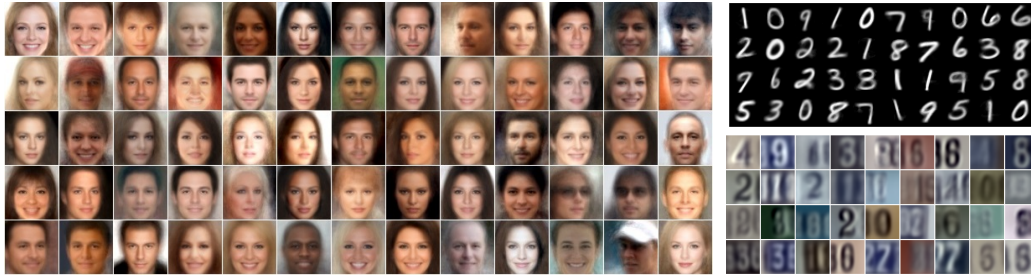

Figure 4: Random samples generated by our MFA model trained on CelebA, MNIST and SVHN

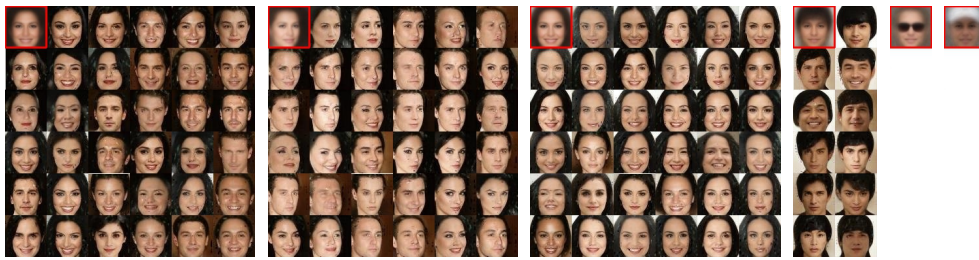

Figure 5: Examples for mode-collapse in BEGAN trained on CelebA, showing three over-allocated bins and three under-allocated ones. The first image in each bin is the cell centroid (marked in red).

Table 1: Bin-proportions NDB/$K$ scores for different models trained on CelebA, using 20,000 samples from each model or set, for different number of bins ($K$). The listed values are NDB – numbers of statistically different bins, with significance level of $0.05$, divided by the number of bins $K$ (lower is better).

| MODEL | $K$=100 | $K$=200 | $K$=300 |
|---|---|---|---|
| TRAIN | 0.01 | 0.03 | 0.03 |
| TEST | 0.12 | 0.07 | 0.08 |
| MFA | **0.21** | **0.12** | **0.16** |
| MFA+*pix2pix* | 0.34 | 0.34 | 0.33 |
| ADVERSARIAL MFA | 0.33 | 0.30 | 0.22 |
| VAE | 0.78 | 0.73 | 0.72 |
| VAE-DFC | 0.77 | 0.65 | 0.62 |
| DCGAN | 0.68 | 0.69 | 0.65 |
| BEGAN | 0.94 | 0.85 | 0.82 |
| WGAN | 0.76 | 0.66 | 0.62 |
| WGAN-GP | 0.42 | 0.32 | 0.27 |

Table 2: NDB/$K$ scores for MNIST

| MODEL | $K$=100 | $K$=200 | $K$=300 |
|---|---|---|---|
| TRAIN | 0.06 | 0.04 | 0.05 |
| MFA | **0.14** | **0.13** | **0.14** |
| DCGAN | 0.41 | 0.38 | 0.46 |
| WGAN | 0.16 | 0.20 | 0.21 |

Table 3: NDB/$K$ scores for SVHN

| MODEL | $K$=100 | $K$=200 | $K$=300 |
|---|---|---|---|
| TRAIN | 0.03 | 0.03 | 0.03 |
| MFA | **0.32** | **0.23** | **0.24** |
| DCGAN | 0.78 | 0.74 | 0.76 |
| WGAN | 0.87 | 0.83 | 0.82 |

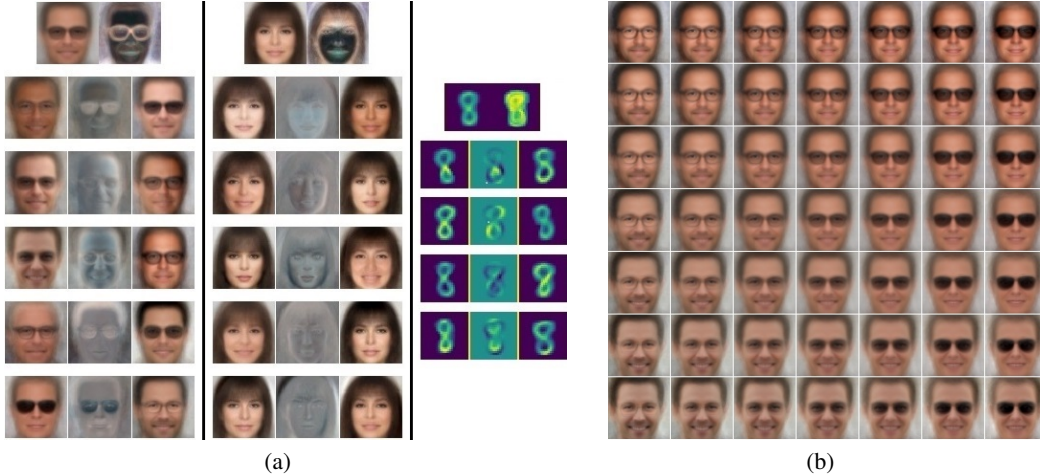

(a)                                                                  (b)

Figure 6: (a) Examples of learned MFA components trained on CelebA and MNIST: Mean image ($\mu$) and noise variance ($D$) are shown on top. Each row represents a column-vector of the rectangular scale matrix $A$ – the learned changes from the mean (showing vectors 1-5 of 10). The three images shown in row $i$ are: $\mu + A^{(i)}$, $0.5 + A^{(i)}$, $\mu - A^{(i)}$. (b) Combinations of two column-vectors $(A^{(i)}, A^{(j)})$: $z_i$ changes with the horizontal axis and $z_j$ with the vertical axis, controlling the combination. Both variables are zero in the central image, showing the component mean.

Our evaluation method can provide visual insight into the mode collapse problem. Fig. 5 shows random samples generated by BEGAN that were assigned to over-allocated and under allocated bins. As can be seen, each bin represents some prototype and the GAN failed to generate samples belonging to some of them. Note that the simple binning process (in the original image space) captures both semantic properties such as sunglasses and hats, and physical properties such as colors and pose. Interestingly, our metric also reveals that VAE also suffers from "mode collapse" on this dataset.

Finally, we compare the models in terms of disentangling the manifold the and ability to perform inference.

It has often been reported that the latent representation $z$ in most GANs does not correspond to meaningful directions on the statistical manifold [6] (see supplementary materia for a demonstration in 2D). Fig. 6(a) shows that in contrast, in the learned MFA model both the components and the directions are meaningful. For CelebA, two of 1000 learned components are shown, each having a latent dimension $l$ of 10. Each component represents some prototype, and the learned column-vectors of the rectangular scale matrix $A$ represent changes from the mean image, which span the component on a 10-dimensional subspace in the full image dimension of $64 \times 64 \times 3 = 12,288$. As can be seen, the learned vectors affect different aspects of the represented faces such as facial hair, glasses, illumination direction and hair color and style. For MNIST, we learned 256 components with a latent dimension of $4$. Each component typically learns a digit and the vectors affect different style properties, such as the angle and the horizontal stroke in the digit 7. Very different styles of the same digit will be represented by different components.

The latent variable $z$ controls the combination of column-vectors added to the mean image. As shown in Fig. 6(a), adding a column-vector to the mean with either a positive or a negative sign results in a realistic image. In fact, since the latent variable $z$ is sampled from a standard-normal (iid) distribution, any linear combination of column vectors from the component should result in a realistic image, as guaranteed by the log-likelihood training objective. This property is demonstrated in Fig. 6(b). Even though the manifold of face images is very nonlinear, the GMM successfully models it as a combination of local linear manifolds. Additional examples in the supplementary material.

As discussed earlier, one the the main advantages of an explicit model is the ability to calculate the likelihood and perform different inference tasks. Fig. 7(a) shows images from CelebA that have low likelihood according to the MFA. Our model managed to detect outliers. Fig. 7(b) demonstrates the task of image reconstruction from partially observed data (in-painting). For both tasks, the MFA model provides a closed-form expression – no optimization or re-training is needed. Both inpainting

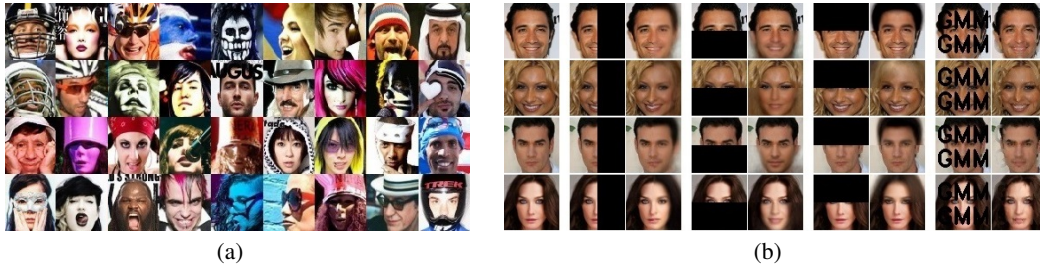

<div align="center">(a)                (b)</div>

Figure 7: Inference using the explicit MFA model: (a) Samples from the 100 images in CelebA with the lowest likelihood given our MFA model (outliers) (b) Image reconstruction – in-painting: In each row, the original image is shown first and then pairs of partially-visible image and reconstruction of the missing (black) part conditioned on the observed part.

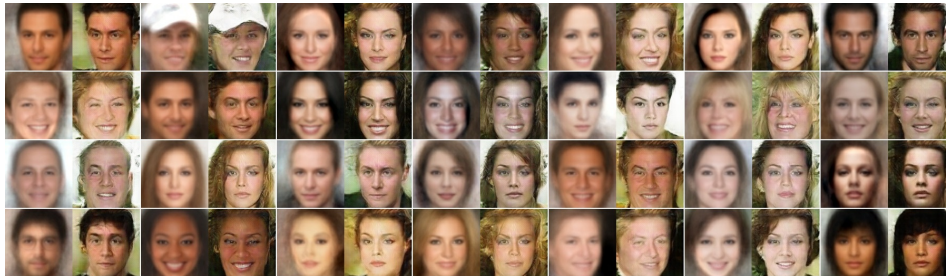

Figure 8: Pairs of random samples from our MFA model, resized to 128x128 pixels and the matching samples generated by the conditional *pix2pix* model (more detailed)

and calculation of log likelihood using the GAN models is difficult and requires special purpose approximations.

## 5   Generating Sharp Images with GMMs

Summarizing our previous results, GANs are better than GMMs in generating sharp images while GMMs are better at actually capturing the statistical structure and enabling efficient inference. Can GMMs produce sharp images? In this section we discuss two different approaches that achieve that. In addition to evaluating the sharpness subjectively, we use a simple sharpness measure: the relative energy of high-pass filtered versions of set of images (more details in the supplementary material). The sharpness values (higher is sharper) for original CelebA images is -3.4, for WGAN-GP samples -3.9 and for MFA samples it is -5.4 (indicating that GMM samples are indeed much less sharp). A trivial way of increasing sharpness of the GMM samples is to increase the number of components: by increasing this number by a factor of 20 we obtain samples of sharpness similar to that of GANs ($-4.0$) but this clearly overfits to the training data. Can a GMM obtain similar sharpness values without overfitting?

### 5.1   Pairing GMM with a Conditional GAN

We experiment with the idea of combining the benefits of GMM with the fine-details of GAN in order to generate sharp images while still being loyal to the data distribution. A *pix2pix* conditional GAN [19] is trained to take samples from our MFA as input and make them more realistic (sharpen, add details) without modifying the global structure.

We first train our MFA model and then generate for each training sample a matching image from our model: For each real image $x$, we find the most likely component $c$ and a latent variable $z$ that maximizes the posterior probability $P(z|x, \mu_c, \Sigma_c)$. We then generate $\hat{x} = A_c z + \mu_c$. This is equivalent to projecting the training image on the component subspace and bringing it closer to the mean. We then train a *pix2pix* model on pairs $\{x, \hat{x}\}$ for the task of converting $\hat{x}$ to $x$. $\hat{x}$ can be resized to any arbitrary size. In run time, the learned *pix2pix* deterministic transformation is

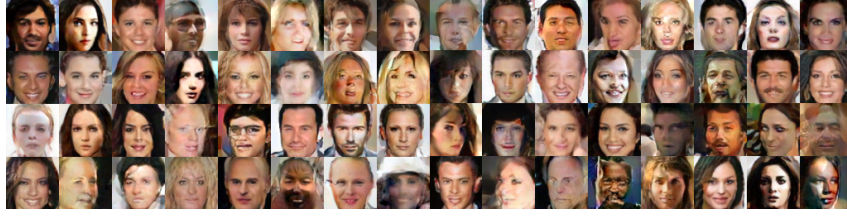

Figure 9: Samples generated by adversarially-trained MFA (500 components)

applied to new images sampled from the GMM model to generate matching fine-detailed images. Higher-resolution samples generated by our MFA+*pix2pix* models are shown in Fig. 8 and in the supplementary material. As can be seen in Fig. 8, *pix2pix* adds fine details without affecting the global structure dictated by the MFA model sample. The measured sharpness of MFA+*pix2pix* samples is -3.5 – similar to the sharpness level of the original dataset images. At the same time, the NDB scores become worse (Table 1).

### 5.2 Adversarial GMM Training

GANs and GMMs differ both in the generative model and in the way it is learned. The GAN *Generator* is a deep non-linear transformation from latent to image space. In contrast, each GMM component is a simple linear transformation ($Az + \mu$). GANs are trained in an adversarial manner in which the *Discriminator* neural-network provides the loss, while GMMs are trained by explicitly maximizing the likelihood. Which of these two differences explains the difference in generated image sharpness? We try to answer this question by training a GMM in an adversarial manner.

To train a GMM adversarially, we replaced the WGAN-GP Generator network with a *GMM Generator*: $x = \sum_{i=1}^{K} c_i(A_i z_1 + \mu_i + D_i z_2)$, where $A_i, \mu_i$ and $D_i$ are the component scale matrix, mean and noise variance. $z_1$ and $z_2$ are two noise inputs and $c_i$ is a *one-hot* random variable drawn from a multinomial distribution controlled by the mixing coefficients $\pi$. All component outputs are generated in parallel and are then multiplied by the one-hot vector, ensuring the output of only one component reaches the Generator output. The Discriminator block and the training procedure are unchanged.

As can be seen in Fig. 9, samples produced by the adversarial GMM are sharp and realistic as GAN samples. The sharpness value of these samples is -3.8 (slightly better than WGAN-GP). Unfortunately, NDB evaluation shows that, like GANs, adversarial GMM suffers from mode collapse and futhermore the log likelihood this MFA model gives to the data is far worse than traditional, maximum likelihood training. Interestingly, early in the adversarial training process, the GMM Generator decreases the noise variance parameters $D_i$, effectively "turning off" the added noise.

## 6 Conclusion

The abundance of training data along with advances in deep learning have enabled learning generative models of full images. GANs have proven to be tremendously popular due to their ability to generate high quality images, despite repeated reports of "mode collapse" and despite the difficulty of performing explicit inference with them. In this paper we investigated the utility of GANs for learning statistical models of images by comparing them to the humble Gaussian Mixture Model. We showed that it is possible to efficiently train GMMs on the same datasets that are usually used with GANs. We showed that the GMM also generates realistic samples (although not as sharp as the GAN samples) but unlike GANs it does an excellent job of capturing the underlying distribution and provides explicit representation of the statistical structure.

We do not mean to suggest that Gaussian Mixture Models are the ultimate solution to the problem of learning models of full images. Nevertheless, the success of such a simple model motivates the search for more elaborate statistical models that still allow efficient inference and accurate representation of statistical structure, even at the expense of not generating the prettiest pictures.

## Acknowledgments

Supported by the Israeli Science Foundation and the Gatsby Foundation.

## Footnotes

[1] Code is available at https://github.com/eitanrich/gans-n-gmms

[2] Flow GAN discusses a full-image GMM, but does not actually learn a meaningful model: the authors use a "GMM consisting of $m$ isotropic Gaussians with equal weights centered at each of the $m$ training points".

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
