[Supplementary Material]

# On GANs and GMMs – Supplementary Material

**Eitan Richardson**
School of Computer Science and Engineering
The Hebrew University of Jerusalem
Jerusalem, Israel
eitanrich@cs.huji.ac.il

**Yair Weiss**
School of Computer Science and Engineering
The Hebrew University of Jerusalem
Jerusalem, Israel
yweiss@cs.huji.ac.il

## 1 Technical Details

### 1.1 NDB Evaluation Method

This section provides additional technical details about the NDB evaluation method.

To define the bin centers, we first perform $K$-means clustering on the training data. To reduce clustering time, a random subset of the data is used (for example, we used 80,000 samples for CelebA). In addition, we sample the data dimension (for CelebA, we used 2000 elements out of $12,288 = 64 \times 64 \times 3$). A standard $K$-means algorithm is then executed, with multiple (10) initializations. Each bin center is the mean of all samples assigned to the cluster (in the full data dimension).

NDB can be performed on the original images or on images divided by the per-pixel data standard deviation (semi-whitened images). In CelebA, due to the large variance in background color, we performed NDB on the images divided by the standard deviation.

In addition to the number of statistically-different bins (NDB), our implementation calculates the Jensen-Shannon (JS) divergence between the reference bins distribution and the tested model bins distribution. If the number of samples is sufficiently high, this soft metric (which doesn't require defining a significance level) can be used as an alternative.

### 1.2 MFA Training

This section provides additional technical details about the training procedure of the MFA model.

Initialization: By default (all reported results), the MFA is initialized using $K$-means. After performing $K$-means, a $Factor Analysis$ is performed on each cluster separately to estimate the initial component parameters. Alternative initialization methods are: random selection of $l + 1$ images for each component. The set of images define the component subspace, and a default constant noise variance is added. Another possible initialization method is $K$-subspaces, in which each component is defined by a random seed of $l + 1$ images, but is then refined by adding all images that are closest to this subspace.

Optimization method: As described, we used Stochastic Gradient Descent (SGD) for training the MFA model. We used the Adam optimizer with a learning rate of 0.0001. All training is implemented using TensorFlow. The training loss is the negative log likelihood. The likelihood of a mini-batch of 256 samples is computed at each training iteration. The gradients (derivatives of the likelihood with respect to the model parameters) are computed automatically by TensorFlow. The model parameters are the mixing coefficients $\pi_i$, the scale matrices $A_i$, the mean values $\mu_i$ and the diagonal noise standard-variation $D_i$. Note that the entire mixture model is trained together.

Hierarchichal training: To reduce training time and memory requirements, when the number of components is large, we used hierarchichal training in which we first trained a model with $K_{root}$ components. We then split each component to additional sub-components using only the relevant

subset of the training data. The number of sub-components depends on the number of samples assigned to the component (larger components were divided to more sub-components). After training all sub-components, we define a flat model from all sub-components by simply multiplying their mixing-coefficients by the root components mixing-coefficients.

## 1.3 MFA Inference – Image Reconstruction

We demonstrate the inference task with image reconstruction – in-painting. Part of an (previously unseen) image is observed and we complete the missing part using the trained MFA model by computing $\mathrm{argmax}_{x_1} P(X_1 | X_2 = x_2)$, where $X_1$ is the hidden part and $X_2$ is the observed part (pixels).

The approach we used to calculate the mode of the conditional distribution is to first find the most probable posterior values for latent variables given the observed variables and then apply these values to the full model to generate the missing variable values. Specifically, we first reduce all component mean and scale matrices to the scope of the observed variables. Using these reduced model, we find the component $\hat{c}$ with the highest responsibility with respect to the observed value (the most probable component). In this component, we calculate the posterior probability $P(z|x)$. The posterior probability is by itself a Gaussian. We use the mean of this Gaussian as a MAP estimate for the posterior $\hat{z}$. Finally, using the original full model, we calculate $x = A_{\hat{c}}\hat{z} + \mu_{\hat{c}}$. Note that it is also possible to sample from the posterior, generating different possible reconstructions.

## 1.4 Measuring Sharpness

Our simple sharpness score measures the relative energy of high-pass filtered versions of a set of images compared to the original images. We first convert each image to a single channel (illumination level) and subtract the mean. To obtain the high-pass filtered image, we convolve the image with a Gaussian kernel and subtract the resulting low-pass filtered version from the original image. We then measure the energy of the original image and of the high-pass version by summing the squared pixel values and then taking the logarithm. We define the image sharpness as the high-pass filtered version energy minus the original image energy. We take the mean sharpness over 2000 images. The method is invariant to scale and translation in the pixel values (i.e. multiplying all pixel values by a constant or adding a constant).

# 2 Additional Figures

This section provides additional examples to the different sections in the main paper.

## 2.1 NDB Evaluation Method

Figure 1: The largest 30 out of 200 bins in the NDB K-means clustering for CelebA. The first image in each row is the bin centroid and the other images are random training samples from this bin.

Figure 2: Similar to Figure 1, but showing the smallest (least allocated) 30 out of 200 bins.

(a)

(b)

Figure 3: The largest (a) and smallest (b) 25 out of 200 bins in the NDB K-means clustering for MNIST (Similar to Figures 1 and 2)

(a)

(b)

Figure 4: The largest (a) and smallest (b) 25 out of 200 bins in the NDB K-means clustering for SVHN (Similar to Figures 1 and 2)

Figure 5: Binning proportion histograms for K=200 on CelebA. Each plot shows the distribution of bin-assignment for 20,000 random samples from the test set and from different evaluated models. For clarity, shown in pairs.

Figure 6: The binning proportions (and therefore the NDB scores) are consistent for different number of bins (100, 200, 300). Note that the same trained MFA and WGAN model is evaluated in all three cases. In all three cases, the distribution of the MFA samples is similar to the reference train distribution (and also has similar NDB as the test samples) while WGAN exhibits significant distortions.

(a)

(b)

(c)

Figure 7: The NDB method detecting mode-collapse in a deep model (VAE-DFC). Note that this bin represents a semantic property (sun-glasses). Note that the first image in each group is the bin centroid. (a) Images from the CelebA training set belonging to one clustering bin (out of 200). (b) From random 20,000 CelebA test images, the ones that are assigned to this bin – 61/20,000=0.03 (similar proportion to the reference binning on the training data). (c) From 20,000 images generated by the VAE-DFC model, the ones that are assigned to this bin – 9/20,000=0.0045.

(a)

(b)

(c)

Figure 8: Another semantic mode-collapse: A mode in the data distribution (both in the training and in the test set) that is significantly under-represented in the generated sample of VAE-DFC. Similar setting as in Figure 7.

(a)

(b)

(c)

Figure 9: In this example, the bin represents a mix of semantic (people with dark skin) and photometric (dark images) latent properties, which have a similar affect on the observed pixels. We argue that the generative model should represent both (i.e. not just the semantic properties).

(a)

(b)

(c)

Figure 10: An example for an over-represented bin in VAE-DFC (0.016 vs 0.007 in the train and test sets)

## 2.2  MFA – Full Image Gaussian Mixture Model

Figure 11: The effect of the number of MFA components on the log-likelihood and quality of represented images. As can be seen, the log-likelihood and reconstruction quality improve quickly with the number of components.

|       (a)       |       (b)       |

Figure 12: Internal representations of generative models (a) GAN learns an elaborate non-linear transformation from latent to data space. z points are on a grid (with larger steps in one dimension) (b) the MFA component centers, direction and added noise

Figure 13: Additional random samples drawn from the MFA model trained on CelebA

(a)

(b)

Figure 14: Additional random samples drawn from the MFA model trained on (a) MNIST and (b) SVHN

Figure 15: Additional examples of learned MFA components trained on CelebA: Mean image ($\mu$) and noise variance ($D$) are shown on top. Each row represents a column-vector of the rectangular scale matrix $A$ – the learned changes from the mean. The three images shown in row $i$ are: $\mu + A^{(i)}$, $0.5 + A^{(i)}$, $\mu - A^{(i)}$.

Figure 16: Combinations of two column-vectors $(A^{(i)}, A^{(j)})$: $z_i$ changes with the horizontal axis and $z_j$ with the vertical axis, controlling the combination. Both variables are zero in the central image, showing the component mean

## 2.3 Generating Sharp Images with a GMM

Figure 17: An illustration of the MFA+*pix2pix* model. The gray curve represents the data manifold and the colored regions, MFA components. Each component resides on a subspace, with added noise. *pix2pix* learns to transform images generated by the MFA model ($X_A \rightarrow X_B$) to bring them closer to the data manifold.

Figure 18: Animated interpolation of the latent vector $z$ plus application of the *pix2pix* model to provide details. **Please see attached animated GIFs.**. Each sample was generated from a single MFA component by traveling along different dimensions of $z$, applying the Factor Analyzer transformation, resizing to $128 \times 128$ and then applying the *pix2pix* model to add details.

Figure 19: Top: Random samples generated from a single component out of 1000 in the MFA model. Middle: Learned *pix2pix* transformation applied to these samples. Bottom: CelebA training images belonging to the selected component (i.e. the component has maximal responsibility value). One can the that the MFA generates diverse images that don't memorize training images and that *pix2pix* adds details while maintaining the original structure.