[Reviews · NeurIPS 2018]

Reviewer 1



Major comments: This work examines GANs by comparing it to a simple mixture of factor analyzers (MFA) using NDB (a score based on sample histograms). The NDB computes the number of statistically different bins where the bins are obtained via Voronoi tessellation on k-means centroids. The key result is that the GMM/MFA is better able to capture the underlying distribution compared to GANs. When the MFA is combined with a pix2pix model, it generates sharp images comparable to the GAN model. Overall, this is a well-written paper with interesting results that question the overall utility of GANs. Its value lies is in its simple, accessible approach and extensive experimental results on a variety of generative models. However, the proposed techniques do not appear substantially novel and I have several concerns/questions that I hope the authors can address: The results hinge on the effectiveness of the NDB. As such, emphasis should be given to establishing the consistency of the measure across initializations and sensitivity to parameters (K, significance-level, # of samples). The authors do state that the method is consistent and robust to initialization (lines 177-180), but some evidence would help bolster confidence in the score, especially since it is a primary contribution of this work. How does the NDB score fare on complex natural image datasets such as CIFAR and/or ImageNet? Some discussion would help clarify the generality and limitations of the score. One attractive aspect of the GAN model is that it learns using an implicit likelihood, as compared to prescribed models (such as GMMs). Mohamed and Lakshminarayanan (2016) give an in-depth discussion of the two approaches. Potentially, the GMM model is able to score high NDB scores since it is optimizing a L2-like score across the pixels? The observation that GMMs can outperform GANs (on standard log-likelihood, MODE/Inception scores), was previously reported in Grover, Dhar and Ermon (2017). Some discussion on what is new in this work would help to clarify the contributions. Sec 4.1: What is the NDB of the random samples from the MFA+pix2pix model? Is it higher than the MFA or comparable? Minor comment: NDB: Is there a good reason for reporting the number of bins rather than proportion? The latter seems more comparable across different K? -- Post-Rebuttal Comments -- Thank you for your responses, particularly for the sensitivity analysis and for clarifying the differences in metrics.

Reviewer 2



Summary The authors compare the performance of Gaussian mixture models (GMMs) and deep generative models (mainly GANs, but also VAEs) for image generation. They use a mixture of factor analysers (MFA) model and train it on several image data sets, and show that this model can faithfully learn the distribution of images. They also propose a new evaluation scheme for generative models, based on Voronoi cells. They use this scheme to compare MFAs, GANs, and VAEs: MFAs perform pretty well, and avoid mode collapse but produce somewhat blurry images. They propose a scheme based on pix2pix to obtain sharper, "GAN-like" images. Globally, the paper is rather thought-provoking and carries an interesting message. Quality Overall, the technical quality of the paper is rather good. Regarding the new evaluation method, I think it makes sense. However, a theoretical assessment (at least superficial) of this method would be welcomed. It is not clear what kind of discrepancy the proposed method measures, while some other popular techniques have clearer interpretations (like the Wasserstein distance, or the test log-likelihood). The MFA part is rather classical but interesting and natural. More details should be added regarding the choice of the dimension of the latent space and the number of components. For example, would it make sense to use different dimensions for the different components. Regarding the experiments, I think the authors do a good job at showing that their metric is a good "mode collapse detector". I also really like that they show how to interpret the latent variables of the MFA model. However, I think that a crucial experiment would be welcomed: the comparison of (both training and test) log-likelihoods for their models and the competitors that allow likelihood evaluation . It is possible to evaluate the likelihood of VAEs using importance sampling, and there are other deep generative models of images, like RNVP (Density estimation using Real NVP, Dinh, Sohl-Dickstein, and Bengio, ICLR 2017) that allow direct likelihood evaluation. Since their model is also trained using maximum likelihood, such experiments would be very insightful. Clarity I think that the paper reads quite well. A few technical points should be clarified (at least in the supplementary material): - What is precisely meant by "a GMM can approximate any density" (l. 120)? With what metric? A citation should back this claim as well as the one stated just after that ("it is easy to construct densities for which the number of gaussian required grows exponentially"). - How is in-painting done exactly? - How is SGD implemented? The authors say that each component is estimated separately (l. 155), could they explain precisely what this means? What is the learning rate? - The relevance of using L2 Voronoi cells should be discussed more. Originality The evaluation method is, to the best of my knowledge, novel. The MFA model is very well known, but using it for image generation is somewhat original (in particular the experiments of Fig. 6b). The pix2pix part introduces also an idea that I've never seen before: first learn roughly the distribution with a technique immune to mode collapse, then refine the samples to produce sharp images. Significance I really like the general message of this paper, which, albeit not surprising, is thought provoking. However, I think that such an empirical paper should have much stronger experiments. In particular, I think that some comparisons with the existing metrics (like FID, test log-likelihood, or estimated Wasserstein distance) should be present in the paper. In particular, it is very surprising not to see likelihood comparisons with other likelihood-based models. Moreover, the sharp samples obtained with pix2pix could be compared to GANs samples under the FID or the Wasserstein metric. Minor comments - In-painting is sometimes spelled in-painting and sometimes inpainting - The title is quite misleading, and the analysis should focus more generally on deep generative models (not only GANs) ------ Post-rebuttal edit ------ I have read the rebuttal, that clarified a few points. My main concern remains the absence of likelihood evaluation for the other techniques. In particular, I think that the likelihood of the VAE trained by the authors should really be compared to the one of the GMM in the final version (for example in the supplementary material). This can be rather easily done using importance sampling, as in many VAE papers. Beyond that, I still think that the overall message is very valuable and that the paper should be accepted.

Reviewer 3



The paper considers the problem of high-dimensional image modeling and generation, and presents a new evaluation scheme, and insightful experiments. The paper compares a Gaussian Mixture Model's performance with that of a state-of-the-art Generative Adversarial Network, and shows that the former is arguably doing a better job of modeling the full data distribution, and while the samples generated by the GMM is realistic, they lack the sharp features of samples from GAN. The paper concludes that perhaps more research could result in models which are easy to train and infer and yet capture the finer details of images like GANs. The paper shows that techniques from the literature, such as factor analysis and the Woodbury matrix inversion lemma, allow training high-dimensional GMMs efficiently. The paper presents a new simple evaluation scheme which does not rely on pre-trained classifiers, but which however do rely on using L2 distance on image pixels. The experiments show that the GMM trained does better on the proposed evaluation scheme compared to GANs. Since the paper relies on the evaluation scheme so much, it would help if the evaluation scheme itself could be evaluated by doing user studies, for instance. The authors also study a method to transform the blurry images produced by the GMM to richer images using pix2pix. This is not central to the central thesis of the paper. In my view, the GMM model is merely an instrument to study a SOTA model, and as the authors conclude as well, not meant as the ultimate solution to the problem of learning generative models. Therefore, I haven't judged this paper by the pretty samples in the supplementary section. In summary, the paper has a very significant topic, proposes an interest idea ("How does a model as simple as GMM compare to SOTA GANs?"), evaluates the idea sufficiently exhaustively, and provides good analysis of the results to inform readers' intuitions.